# Polycrystal Simulation of Texture-Induced Grain Coarsening during Severe Plastic Deformation

**DOI:** 10.3390/ma13245834

**Published:** 2020-12-21

**Authors:** Chi Zhang, Laszlo S. Toth

**Affiliations:** 1School of Materials Science and Engineering, Dalian University of Technology, Dalian 116024, China; zhangchi@dlut.edu.cn; 2Laboratory of Excellence on Design of Alloy Metals for Low-mAss Structures (‘LabEx DAMAS’), Université de Lorraine, F-57070 Metz, France; 3LEM3, CNRS, Arts et Métiers ParisTech, Université de Lorraine, F-57070 Metz, France

**Keywords:** grain coarsening, crystallographic texture, polycrystal plasticity modeling, severe plastic deformation (SPD)

## Abstract

During severe plastic deformation (SPD), there is usually extended grain fragmentation, associated with the formation of a crystallographic texture. The effect of texture evolution is, however, coarsening in grain size, as neighbor grains might coalesce into one grain by approaching the same ideal orientation. This work investigates the texture-induced grain coarsening effect in face-centered cubic polycrystals during simple shear, in 3D topology. The 3D polycrystal aggregate was constructed using a cellular automaton model with periodic boundary conditions. The grains constituting the polycrystal were assigned to orientations, which were updated using the Taylor polycrystal plasticity approach. At the end of plastic straining, a grain detection procedure (similar to the one in electron backscatter diffraction, but in 3D) was applied to detect if the orientation difference between neighboring grains decreased below a small critical value (5°). Three types of initial textures were considered in the simulations: shear texture, random texture, and cube-type texture. The most affected case was the further shearing of an initially already shear texture: nearly 40% of the initial volume was concerned by the coalescence effect at a shear strain of 4. The coarsening was less in the initial random texture (~30%) and the smallest in the cube-type texture (~20%). The number of neighboring grains coalescing into one grain went up to 12. It is concluded that the texture-induced coarsening effect in SPD processing cannot be ignored and should be taken into account in the grain fragmentation process.

## 1. Introduction

The grain size is one of the key factors determining the performance of metals and alloys. In general, grain refinement improves the strength, which can be explained by the empirical Hall-Petch relation. One efficient way to reduce grain size is to apply severe plastic deformation (SPD) on the material, a technique, which gained a lot of attentions during the last two decades, since it can produce submicron- to nanometer sized- grains in a wide variety of materials at relative low temperatures [1,2,3]. The most studied SPD techniques are equal-channel angular pressing (ECAP) [4,5], high-pressure torsion (HPT) [6,7], and accumulative roll bonding (ARB) [8], whereas others are emerging [9]. These SPD techniques present attractive application potentials as they can enhance the strength of conventional metallic materials dramatically [10].

A series of models were developed to simulate the grain fragmentation during SPD process [11,12,13,14,15]. These models show reasonable agreements on the grain size variations and texture evolution. However, very less attention is paid on the phenomenon of possible grain coarsening during SPD. The first and only study on grain coarsening was presented by Gu et al. [16], who observed this phenomenon during room temperature sheet-ECAP in a low carbon steel after four passes; the initial grain size increased from 3.75 to 9.57 μm (in number-weighted average, the same in area-weighted average was 8.96–24.2 μm). They explained the phenomenon by the evolution of the crystallographic texture, which takes place during large strain deformation. Namely, if neighbor grains are approaching the same ideal orientations during deformation, the disorientation between them is diminishing progressively. Then, part of the high angle grain boundaries can transform to low angle grain boundaries, causing the coalescence of neighbor grains into larger grains. Undoubtedly, this coarsening effect would change the microstructure morphologies and the grain size distribution.

Generally, a steady state develops in the microstructure in SPD at extreme large strains with the average grain size remaining constant [17,18,19]. This requires that the grain fragmentation process should be balanced by some coarsening effects. It is proposed that grain fragmentation is balanced by the occurrence of a continuous dynamic recrystallization process (CDRX) in this stage, by grain boundary migration [20]. Nevertheless, it is also reasonable to assume that a certain number of grains can coalesce together because their misorientation can decrease due to texture formation. It is important to have an estimation on the extent of this texture-induced coalescence effect, for understanding the steady state of the grain fragmentation process in SPD.

The simulation of grain-coalescence requires a topological approach. Gu et al. performed a 2D simulation in Ref. [16] and found that about 17% of the grain population was concerned by coarsening during sheet-ECAP. It is, however, necessary to perform 3D analyses in order to obtain quantitative results. Several methods have been established for studying 3D polycrystal geometry; i.e., cellular automaton (CA), Monte Carlo, Voronoi, etc. [21,22,23]. Among these models, the CA method models the entire 3D grain geometry based on voxels, which works well with the discrete spatial detection [24]. CA is especially versatile and computationally efficient. Moreover, the 3D grain boundary morphology generated by CA is a convex surface and the grain size distribution follows Gaussian distribution, which are realistic properties for representing a polycrystal. Based on the spatial relations of grains in the polycrystal aggregate, it is possible to capture the grain coarsening due to texture evolution.

The subject of the present work is to establish 3D polycrystalline aggregates using a 3D CA method. The texture evolution was simulated for simple shear deformation mode using the Taylor polycrystal model. The coalescence of grains induced by the texture evolution was tracked by detecting the disorientations between neighbor grains. The material volume that was concerned by grain coalescence depended on the initial texture; it was the most significant for an initial shear texture, where it went up to near 40% in a shear strain of about 4.

## 2. The Simulation Scheme

### 2.1. Generation of a 3D Polycrystal Structure Using Cellular Automaton

A 3D CA algorithm was developed to generate 3D polycrystal aggregates. A cubic space was divided into voxels and a large number of grain-sites were generated randomly in this space (typically 10,000). Then, the nuclei were grown until they occupied the whole space using a probability switching rule. Periodic boundary conditions were applied to the cube for simulating an infinite system. During the growth procedure, the Moore neighborhood approach was used; it considers the nearest 6 voxels and the second-nearest 12 voxels in the state switching rule. Grain IDs were assigned to each grain in order to track the grains; all voxels belonging to the same grain had the same grain ID. Finally, crystallographic orientations were assigned to the grains according to the grain IDs. Each voxel in the polycrystal had 7 variables in total; they were position: *x*, *y*, and *z*; grain ID; and the three Euler angles: φ_1_, Φ, and φ_2_.

In this work, the representative volume elements (RVE) of the polycrystal were constructed at three resolution levels: 50 × 50 × 50, 100 × 100 × 100, and 150 × 150 × 150 voxels along the *x, y, z* axes, in each case containing 10,000 grains. For the highest resolution case, the average number of voxels that composed one grains was 338 (150 × 150 × 150/10,000). The voxel size was set as 1 μm × 1 μm × 1 μm. It should be noted that all grain size statistics data in the following are based on this voxel size, however, the fraction and frequency data are independent of the voxel size. To investigate the effects of initial texture on the texture-induced coarsening, three kinds of typical textures, including random texture, cube-type texture, and SPD type shear texture, were assigned to the polycrystal. The initial random texture was generated using the ATEX software [25]. The cube-type and the SPD textures were obtained from experimental data, discretized to 10,000 orientations using ATEX. ATEX is using the cumulative ODF-based technique for discretization, which is described in detail in Ref. [26].

### 2.2. Simulation Procedure

The simulation work consisted of two main parts: first the simulation of the evolution of the texture, and second, the 3D analysis of the orientation relationship between neighboring voxels for detecting their coalescence tendency. Figure 1 shows the flow chart of the simulation stages. The initial textures were discretized to 10,000 orientations and updated during the imposed deformation using the Taylor polycrystal plasticity model (see the Appendix A for the used Taylor approach). In this stage, only the orientations were calculated along the imposed deformation without considering the spatial distribution and neighbor effects of grains in the polycrystal. The 12 {111} <110> slip systems were considered for dislocation slip for face-centered cubic (FCC) crystals. Hardening was not taken into account in the simulation. This can be justified by several arguments, i.e., i. during large strains, the hardening saturates and ii. as the objective is to obtain orientation evolution, isotropic hardening would not influence it. Thus, the results obtained are valid for isotropic hardening, without actually explicitly introducing it into the simulation. The strain rate sensitivity index in the viscoplastic slip law was taken as m = 0.05, which is a suitable usual value for texture simulations by crystal viscoplasticity [27]. As for deformation mode, the simple shear deformation was selected for studying the grain coalescence process, by applying a macroscopic velocity gradient with the only nonzero component $L_{12}=\;\dot{\gamma }$, corresponding to the shear rate imposed on plane 2 in direction 1. The strain increment for the deformation was taken as 0.05 in simple shear. After the crystal plasticity analysis, the simulated orientations at selected strains were output and assigned to the CA polycrystal model for microstructure analysis in the following section.

### 2.3. Microstructure Analysis

The 3D topology of the polycrystal and its orientation evolution during deformation were tracked during the simulation. A grain coalescence detection subroutine was developed to count the volumes that were reallocated between neighbor grains. This was done when the disorientation angle between two voxels belonging to original neighbor grains was less than 5°. After the coalescence detection process, the grain size distributions were calculated in both number-weighted and volume-weighted versions, as well as the average grain sizes.

The initial and the deformation-induced textures of the aggregate were calculated and plotted in pole figures using the ATEX software. For visualization of the 3D orientation data, the orientations of the initial and the deformed state of the polycrystal were output to a set of data with positions and corresponding Euler angles for input into the DREAM.3D software [28]. It permitted to calculate the grain size and disorientation distributions. After that, the ParaView software was used to display the generated microstructure [29].

## 3. Simulation Results

### 3.1. Effect of Topology and Number of Voxel Division

To examine the effects of topology of the polycrystal orientation distribution as well as the mesh size on the texture-induced grain coarsening, the same initial texture was used for the three mesh sizes. First, a random texture was generated using the ATEX software containing 10,000 grains, with grain sizes varying between 4 and 14 μm, see the grain size distribution in Figure 2 for the highest voxel division case. (The grain size was calculated as the diameter of a sphere having the same volume as an irregular-shaped grain.) Then, the grains were labeled by their grain IDs and were distributed randomly in the cubic space based on the CA nucleation algorithm. A visualization of the three kinds of polycrystals is presented in Figure 3, together with their crystallographic textures, in {111} pole figures. As can be seen, the textures are very random in each case, approaching closely the random intensity value (i.e., 1.0). They are, however, not identical, because the assignment process of grains to their corresponding voxels does not permit to reproduce exactly the same relative volume fractions of the grains. The differences are due to the different voxel sizes as well as the nucleation procedure used to fill the grain volumes in the 3D space.

The orientations of the grains were updated using the Taylor crystal plasticity model for the three RVE cases up to a shear strain of 8. Then, the grain coalescence analysis was performed. The results obtained are presented in Figure 4. With the increasing shear strain, the coarsening fractions increased, confirming the occurrence of texture-induced grain coarsening. The nonzero values in the initial states mean that the disorientation angles between some neighbor voxels with different grain IDs happened to be less than 5° during the orientation assignment. As can be seen, significant volume of the polycrystal was concerned by the coalescence effect; up to about 30% at a shear strain of 6. The differences of the obtained volume fractions among the three RVE cases appear very small. This means that the 3D polycrystal topology has limited effect on the average simulation results. It will be shown in the next section that the grain coarsening effect is mainly due to the initial texture and its subsequent evolution.

Since the polycrystalline topology and voxel condition have little effects, the RVE with 150 × 150 × 150 voxels was chosen to simulate the texture-induced grain coarsening effect. It was studied for three different initial textures, i.e., random, cubic-type, and initial shear texture.

### 3.2. Texture-Induced Grain Coarsening in 3D Polycrystals with Random Initial Texture

Some main results were already presented in the preceding section for the random case, here, we examine the texture evolution and the grain size variations. Figure 5 presents the {111} pole figures of the texture evolution during shear deformation of the polycrystal with random initial texture. These pole figures present typical shear textures of FCC polycrystals, corresponding to the Taylor model [27,30]. The texture evolution is relatively fast, i.e., at already a shear strain of 1, the shear texture is present with both of its fiber components, the A and B fibers. At larger strains, the evolution consists in increasing the texture intensity at selected ideal orientations along the fibers, so the fiber aspect is decreasing. There is also noticeable tilt of the texture from its ideal position at the shear of 1, which is decreasing at further strain, and there is even a slight tilting in the shear direction at the largest strain. These texture effects were already interpreted with the help of the rotation field and the rigid body rotation in Ref. [31].

A grain coalescence detection procedure was carried out to track the grain coarsening events during deformation. The obtained results are listed in Table 1. Note that during the random initial texture assigning procedure, some neighbor grains happened to coalesce already at zero strain. Importantly, 130 prior grain produced coalescence forming 64 new grains. This resulted in the grain number of the initial polycrystal to be 9934 instead of 10,000. During plastic deformation, the disorientation angles between some adjacent grains decreased below 5°. Therefore, the grain number of the polycrystal decreased with increasing shear strain as listed in Table 1. The accumulated coalescing volume fraction was also calculated. It increased constantly at the beginning and then reached a saturation state from about shear strain of 4–8. At the shear of 6, the coarsening volume fraction reached a maximum with 29.7%. This means that about 30% volume fraction of the polycrystal was participating in the grain coarsening effect induced by the texture evolution. The changes in average grain sizes, however, were relatively small, see Table 1. In number fraction, the initial average grain size of 8.48 μm increased up to 8.82 μm, while in volume fraction, from 9.00 to 10.38 μm. The grain size distribution changed only slightly (Figure 6); the intensity became higher in the larger grain size domain and the maximum intensity decreased.

### 3.3. Cube-Type Initial Texture

Cube-type textures are frequent in FCC materials. Here, we took an experimental texture measured by electron back-scatter diffraction (EBSD) in an aluminum sheet. The texture was discretized to 10,000 orientations and then assigned to the 3D polycrystal model established in Section 2. Figure 7 displays the 3D initial texture. As can be seen, a near-cube is the major component but there exist other orientations as well.

Figure 8 shows the texture evolution during simple shear deformation for the polycrystal with cube-type initial texture. Similar to the texture evolution for the random case (see Figure 5), the grain orientations approach the preferred orientations quickly at the beginning of shearing. A difference is that the texture intensity is smaller. Another difference is that the two fiber textures are less continuous; at higher strains, the texture is composed mainly of individual components (mostly C and A/$\;\bar{A}$).

The results of the grain coalescence analysis are listed in Table 2. Compared with the random case, the coarsening tendency decreased for the cube-type texture. The maximum was at a shear strain of 4, with slightly more than 20% volume, after which the coalesced volume decreased.

### 3.4. Shear Texture

Most of the SPD processes produce shear textures in the material with very fine recrystallized grains. Here, we have chosen an experimental shear texture obtained in plastic flow machining of Al1050 [32], see Figure 9 for the initial texture as well as its evolution during further shearing. Note that the initial shear texture was rotated to a new reference system in which the shear plane is parallel to axis *X* for further simple shear simulation. The initial shear texture appears with strong C and A1* components. At a shear of 1, a continuous B fiber appears which progressively changes into some individual main components (C, A, and $\;\bar{A}$). The significant increase in texture intensity is due to the fact that the Taylor plasticity model does not consider the interaction between neighbor grains, the orientations can rotate to the ideal orientations rapidly.

The grain coalescence analysis results of the polycrystal with shear texture are listed in Table 3. The coarsening fraction reached 38, 5% at the shear strain of 4.

## 4. Discussion

The volume fraction of coalescing grains is plotted in Figure 10a as a function of shear strain for the three initial textures considered in Section 3 above. It clearly indicates that the polycrystal with shear initial texture has the highest coarsening tendency, followed by the random and cube-type textures. It is also apparent that the volume fraction of coalescing grains is related to the strength of the texture. Figure 10b shows the texture index calculated by the ATEX software as a function of the shear strain. The texture index is obtained as the integral of the square of the texture function, which can be used to estimate the sharpness of the texture [25,29]. As can be seen, the evolution of the texture index has similar features to the evolution of the coalescing volumes. This is expected as the texture index is increasing by the strengthening of the ideal components, so grain orientations approach each other more and more as a function of strain.

It is also apparent from Figure 10 that after the coalescing volume reached a maximum, there was a decrease in the volume. This is actually due to a grain splitting mechanism, which can also take place due to texture evolution. This effect concerned grains that first coalesced, then, at a later deformation stage, the same neighbor-voxels that were responsible for the coalescence evolved on different orientation trajectories, so the orientation difference was increasing between them. The number of grains that were concerned by the splitting mechanism can be seen in Table 1, Table 2 and Table 3; it started at a shear strain of 6 for the random texture and at 4 for the cubic-type and shear textures. So this splitting effect is also a texture effect; its origin resides in the convergent/divergent nature of the rotation field in Euler space. Namely, it has been shown that in simple shear the ideal orientations are positioned between a convergent and divergent rotation field, and that grain orientations can slowly cross from the convergent side to the divergent part of the rotation field [33,34,35]. Figure 11 shows a selected splitting event between two grains. They were disoriented by 11° at the initial state, coalesced at shear of 1, then split at the shear of 3. By further straining, the disorientation between these two grains went up to 57°, which is almost the maximum orientation distance between two cubic crystals (the maximum is 62.5° [36]).

The frequency of the prior grains coarsening together into 754 grains at the shear strain of 2 was analyzed, see Figure 12 (note the logarithmic scale for the frequency). It infers that 66.4% of the coarsening grains (501 grains) are two-grain events. The maximum number of grains that one coarsening grain could contain reached 12. The frequency shows a near exponential decay with the numbers of prior grains that one coarsening grain can contain. The frequency analysis at larger strains showed similar tendencies, except that the coalescence frequency of more than two grains into one was increased.

It is also interesting to examine the morphological features of the coalescing grains; some selected examples are displayed in Figure 13. Note that the color code in this figure corresponds to the Euler angles by attributing the (φ_1_, Φ, and φ_2_) values to the R, G, and B components of colors, respectively. As can be seen in Figure 13, when large number of initial grains coalesced into one grain, the shape could be quite particular; it can be elongated in a certain direction. This effect might produce large difference in the shapes of grains with respect to their shape evolution corresponding to the uniform Taylor deformation condition. Thus, grains can be less, or more elongated, than expected from the uniform deformation mode.

Figure 14a shows the part of the grain population that coalesced into one grain at a shear strain of 2. The corresponding texture is depicted in a {111} pole figure in Figure 14b. The colors correspond to the shear direction (axis x) according to its position in the standard stereographic triangle. As can be seen, most of the grains have near green color, which means that they were oriented with their <110> axis near to the shear direction. Such orientations are part of the ideal B fiber (with the main components A, B, $\;\bar{B}$, and C) and form the large part of the pole figure in Figure 14b. The intensity of the texture in this pole figure is much larger than the texture of the whole population (compare to Figure 9d), which further confirms that the coarsening part mainly comes from those grains that are rotated near to the ideal orientations during deformation.

It is also important to compare the 3D and 2D simulation approaches. Gu et al. [16] reported about texture-induced grain coarsening in a severe plastic deformed low carbon steel and also modelled the coarsening tendency successfully using a 2D approach. They speculated that the effect of coalescence coarsening should be about three times larger when simulations are done in 3D instead of 2D. It was possible to carry out such comparison quantitatively in our approach. For this purpose, we selected a 2D slice from the polycrystal model. The case of the random initial texture was chosen, at the shear strain of 2. Figure 15a shows the polycrystal in the deformed state, its coarsening part is displayed in Figure 15b, and the top surface of the cube is presented in an inverse pole figure (IPF) map in Figure 15c. The same top surface is then plotted twice; in Figure 15d, all grains that made coalescence in 3D are shown in color, while in Figure 15e, only the grains that are coalesced in 2D, within the selected top surface of the cube. (Note that since periodic boundary conditions were applied to the polycrystal model, the voxels in the top surface also had neighbors on the other surface.) The 3D coarsening fraction was 18.2%, while the 2D calculation on the same population gave only 4.5%. The ratio of the two fractions is about 4, which is even higher than the number proposed by Gu et al. Clearly, the grains have many more number of neighbors in 3D, moreover, the coalescence can also “propagate” in 3D, while its propagation is very limited in 2D. Therefore, the coalescing volume fraction is much higher in 3D than in 2D.

The present work examined the texture-induced coalescence/splitting of grains during large plastic strain. Grain refinement by fragmentation was not included in the simulation work. A 3D topological polycrystal model was constructed and coupled to the Taylor polycrystal model. It has been found that the texture-induced coalescence can concern large volume fraction of the polycrystal; up to about 40% during a shear strain of about 4, depending on the initial texture. This is a significant effect, which should be considered in SPD simulations. A continuation of this kind of modeling could be the simultaneous simulation of grain fragmentation and grain coarsening processes and to analyze the relative contributions of the occurrence of fragmentation and coalescence to the overall grain size variation. It is well documented by SPD experiments that the average grain size decreases quickly at the initial deformation stage; then the rate becomes low with increasing strain and even zero at extremely large strains [37,38]. It is then probable that grain coarsening by texture-induced coalescence becomes relevant at a given stage of the grain refinement process. Concerning the Taylor polycrystal model, simulations can be also done using the more sophisticated self-consistent modeling. Nevertheless, similar results are expected because the coalescence effect is operating near the ideal orientations and they are the same for both the Taylor and self-consistent modeling.

## 5. Conclusions

The grain-coarsening effect induced by the evolution of the crystallographic texture was modelled in the present work. The approach consisted in establishing a 3D polycrystal aggregate with large number of grains (10,000). Each grain was composed of voxels (typically several hundred) and its orientation evolution was followed by the Taylor crystal plasticity model. Simple shear deformation was considered as this is the most frequent in SPD experiments. The coalescence events of neighbor grains were detected when the disorientations between neighboring voxels of adjacent grains became less than 5°. Our study resulted in the following main conclusions:The texture-induced grain size coarsening effect can happen to a large volume fraction of grains of a polycrystal; up to ~40% during a shear strain of about 4.The coalescing volume depends on the initial texture. Among the three kinds of initial textures considered, it was the most relevant for shear textures (~40%), followed by the random case (~30%) and was the smallest for cube-type initial texture (~20%).The coalescence can be continued by splitting of the grains that coalesced together, which is an effect of the convergent/divergent nature of the rotation field around the ideal orientations in orientation space, typical for simple shear deformation.The coalescence frequency is about 4 times larger in 3D polycrystals compared to modelling in 2D.

## Figures and Tables

**Figure 1 materials-13-05834-f001:**
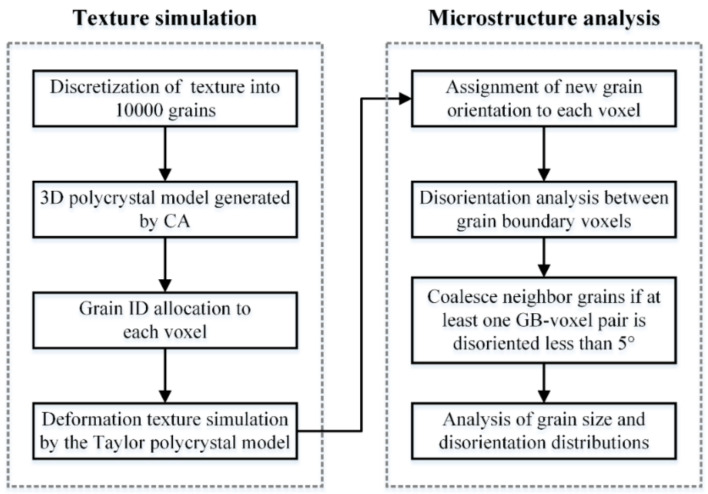
Flow chart of the stages of the simulation of texture-induced grain coarsening in a 3D polycrystal.

**Figure 2 materials-13-05834-f002:**
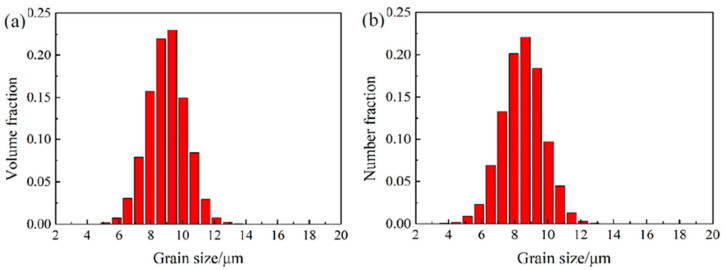
Grain size distribution of a random 3D polycrystal in a representative volume elements (RVE) of 150 × 150 × 150 voxels for 10,000 grains: (**a**) volume fraction and (**b**) number fraction.

**Figure 3 materials-13-05834-f003:**
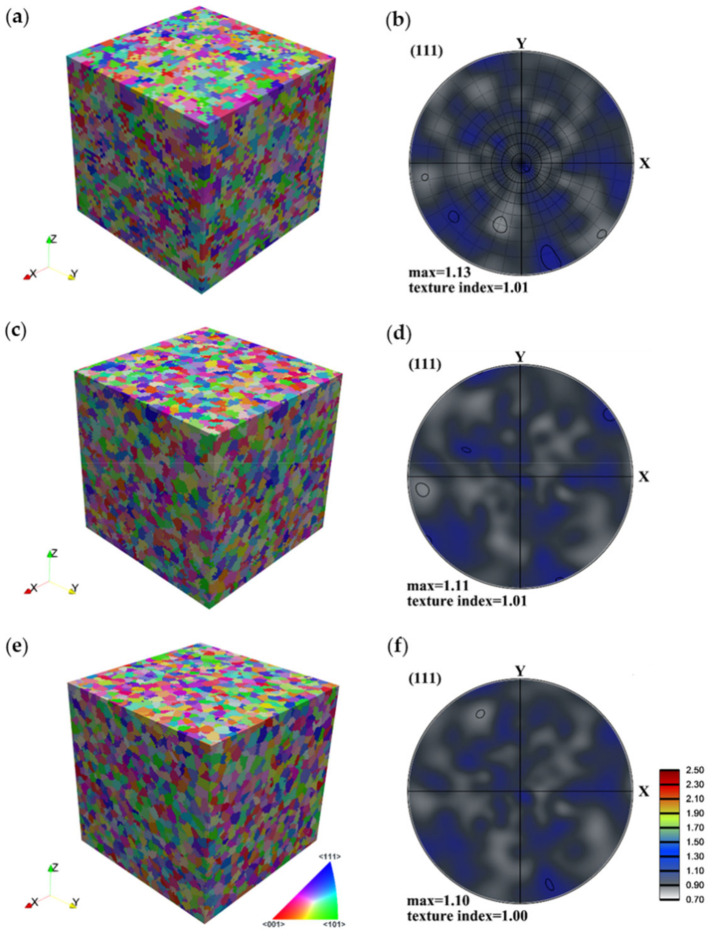
The 3D polycrystal models constructed using the same 10,000 randomly oriented grains: (**a**,**b**) 50 × 50 × 50 voxels, (**c**,**d**) 100 × 100 × 100 voxels, and (**e**,**f**) 150 × 150 × 150 voxels.

**Figure 4 materials-13-05834-f004:**
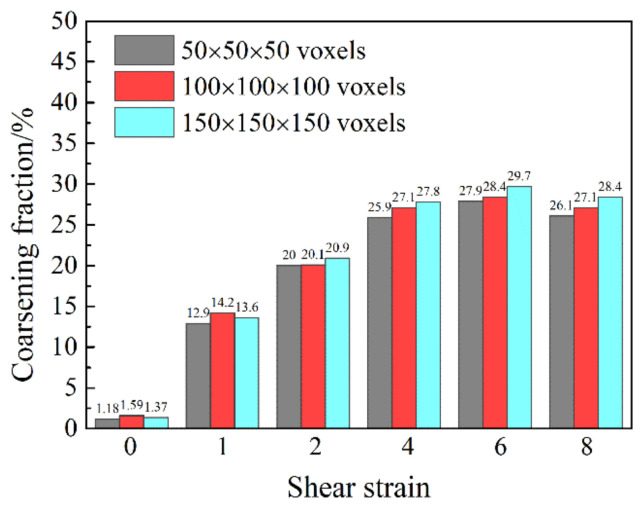
Dependence of the fraction of grain coalescence on the number of voxels constituting the 3D polycrystal as a function of strain.

**Figure 5 materials-13-05834-f005:**
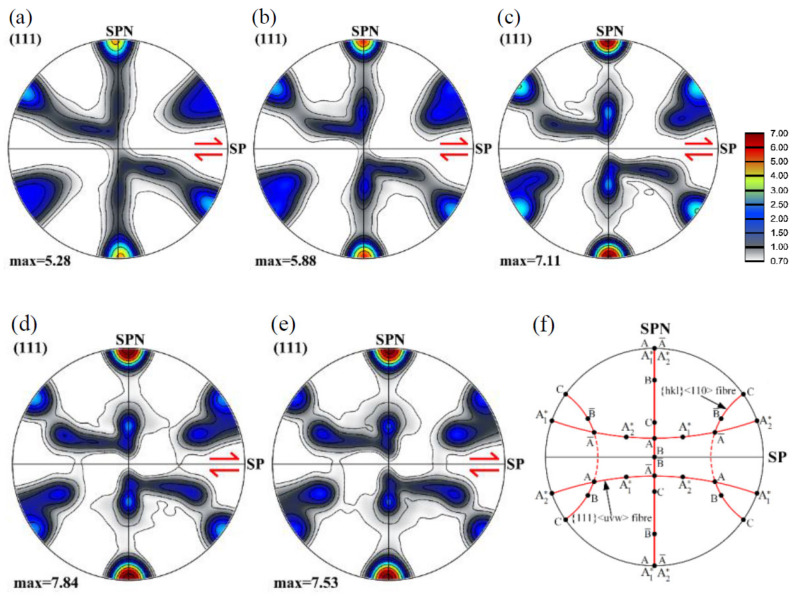
Texture evolution in {111} pole figures of a 3D polycrystal with random initial texture at shears of 1 (**a**), 2 (**b**), 4 (**c**), 6 (**d**), and 8 (**e**), and the ideal orientations of face-centered cubic (FCC) shear textures in {111} pole figure (**f**). (SP indicates the shear plane, and the arrow is orientated in the shear direction; SPN is the shear plane normal).

**Figure 6 materials-13-05834-f006:**
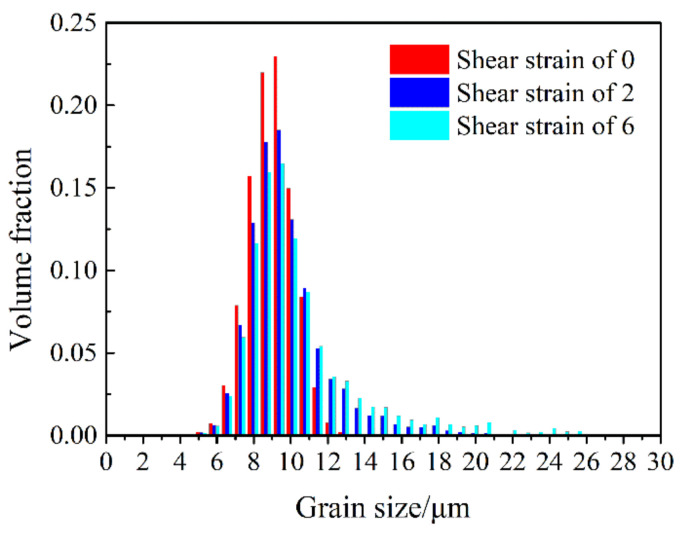
Variation of the grain size distribution as a function of shear strain, starting with random texture.

**Figure 7 materials-13-05834-f007:**
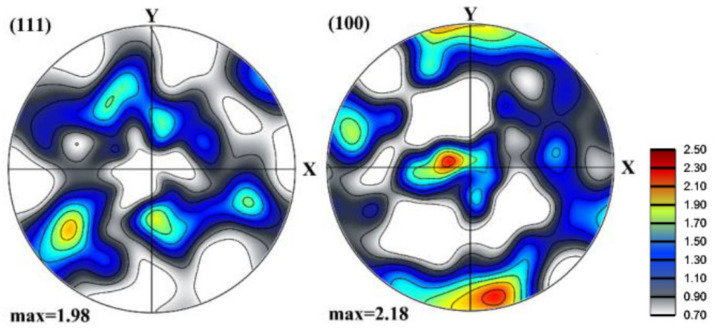
Texture of the 3D polycrystal with cube-type initial texture (discretized from a texture of an aluminum sheet).

**Figure 8 materials-13-05834-f008:**
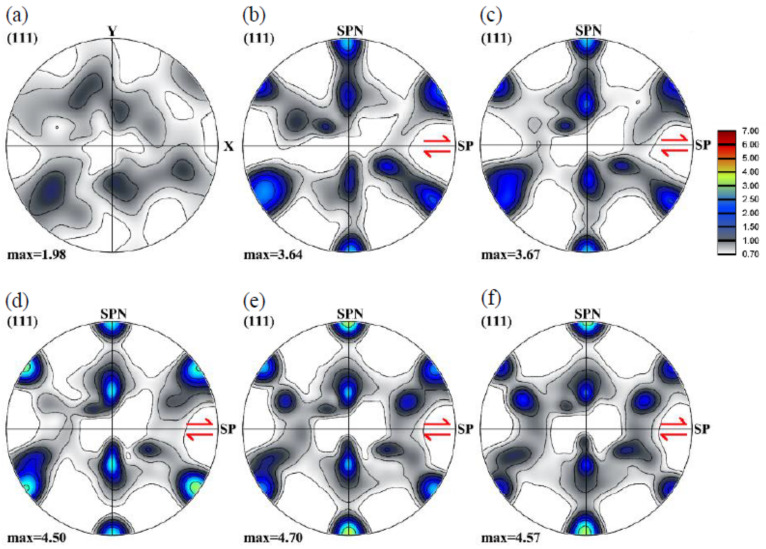
Texture evolution in {111} pole figures of a 3D polycrystal with cube-type initial texture at shears of 0 (**a**), 1 (**b**), 2 (**c**), 6 (**d**), 6 (**e**), and 8 (**f**). (SP indicates the shear plane, and the arrow is orientated in the shear direction; SPN is the shear plane normal).

**Figure 9 materials-13-05834-f009:**
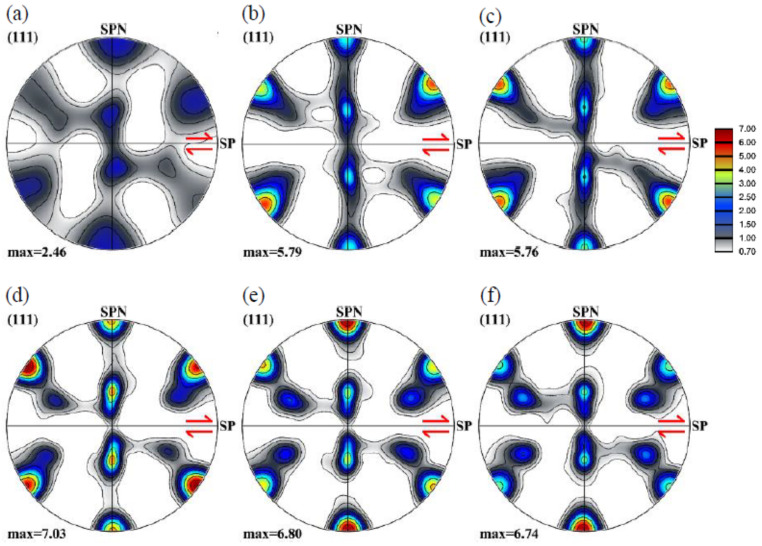
Texture evolution in {111} pole figures of a 3D polycrystal with shear-type initial texture (a) at shears of 0 (**a**), 1 (**b**), 2 (**c**), 6 (**d**), 6 (**e**), and 8 (**f**). (SP indicates the shear plane, and the arrow is orientated in the shear direction; SPN is the shear plane normal).

**Figure 10 materials-13-05834-f010:**
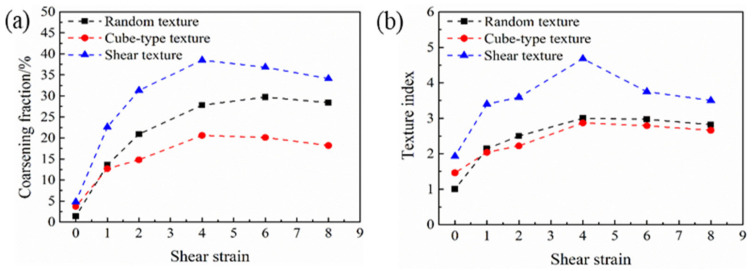
Simulated volume fractions of coalescing grains (**a**) and texture index (**b**) for three kinds of initial textures in simple shear as a function of strain.

**Figure 11 materials-13-05834-f011:**
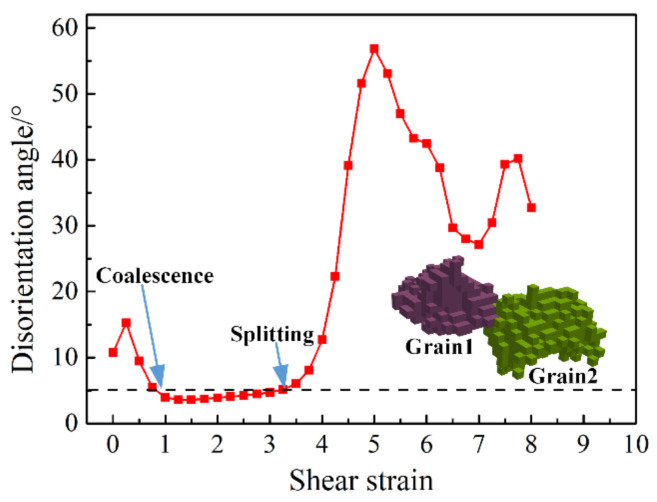
A selected example for coalescing and splitting between two grains displayed by the evolution of their disorientation.

**Figure 12 materials-13-05834-f012:**
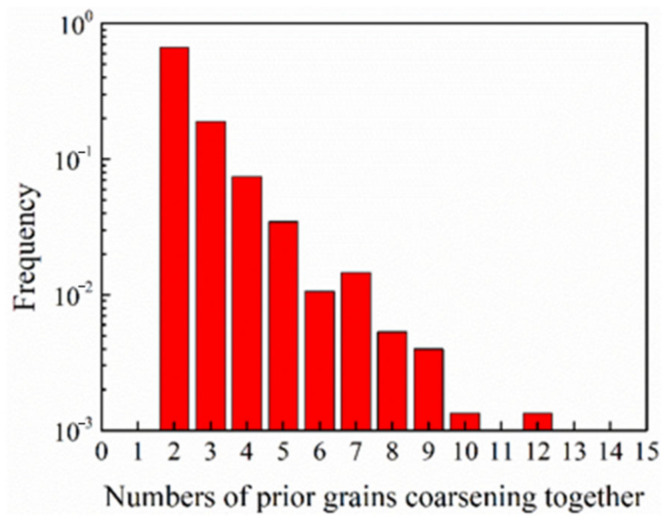
The frequency of number of prior grains coalescing into one grain at a shear strain of 2.

**Figure 13 materials-13-05834-f013:**
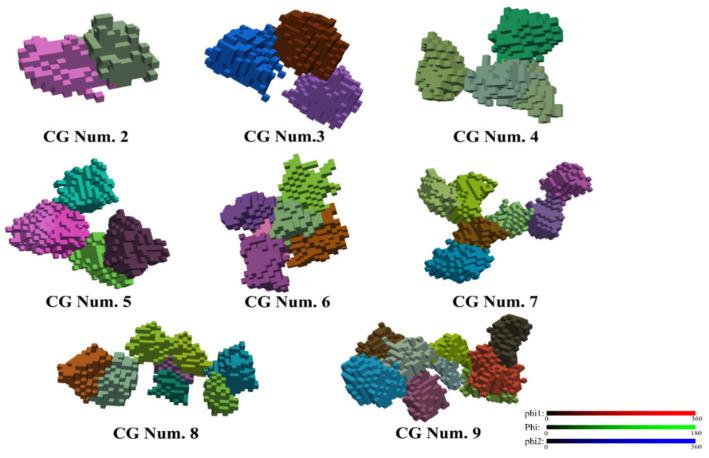
Typical morphologies for grain coalescence at a shear strain of 2. Note that the deformation-induced grain shape change is not displayed; the grains are represented in their initial form. The colors refer to the Euler angles. CG Num. indicates the number of coalescing initial grains.

**Figure 14 materials-13-05834-f014:**
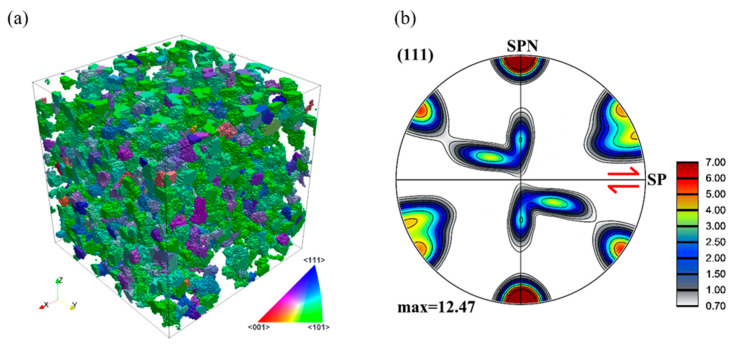
The texture-induced grain coarsening part of the polycrystal displayed by inverse pole figure (IPF) 3D mapping in the real space (**a**) and its corresponding texture at a shear strain of 2 (**b**). The starting texture was random.

**Figure 15 materials-13-05834-f015:**
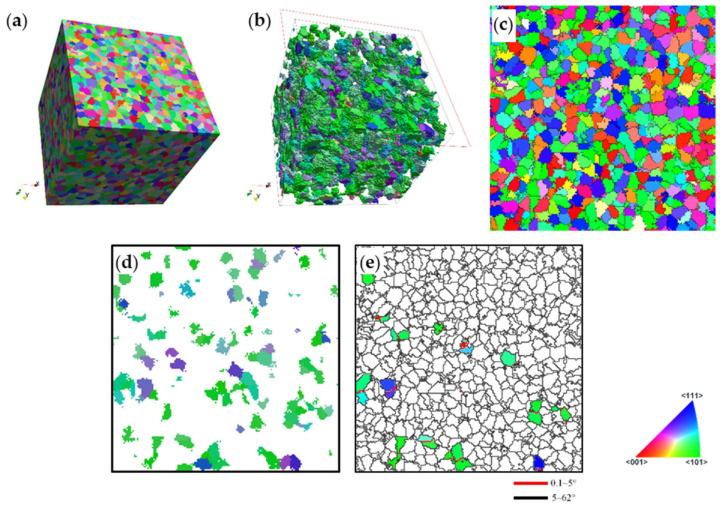
Comparison between 3D and 2D modeling of the texture-induced grain coalescence. (**a**) The polycrystal of initially random orientation distribution after a shear strain of 2, presented in the initial 3D cube. (**b**) The coarsening population part of the polycrystal. (**c**) The IPF map of the top surface of the polycrystal. (**d**) Grains that are coalescing with others being in the top surface in 3D, and the same in (**e**) in 2D.

**Table 1 materials-13-05834-t001:** Grain coalescence analysis results for the polycrystal with random initial texture.

Shear Strain	0	1	2	4	6	8
Grain number	9934	9250	8763	8238	8095	8201
Average grain size (number-weighted)/μm	8.48	8.62	8.72	8.81	8.84	8.82
Average grain size (volume-weighted)/μm	9.00	9.38	9.76	10.27	10.45	10.38
Number of new grains by coalescence	64	555	754	897	937	926
Number of coalescing initial grains	130	1305	1991	2659	2842	2725
Coalescing volume fraction	1.4%	13.6%	20.9%	27.8%	29.7%	28.4%

**Table 2 materials-13-05834-t002:** Grain coalescence analysis results for the polycrystal with cube-type initial texture.

Shear Strain	0	1	2	4	6	8
Grain number	9828	9320	9153	8706	8754	8914
Average grain size (number-weighted)/μm	8.50	8.60	8.63	8.70	8.70	8.68
Average grain size (volume-weighted)/μm	9.06	9.35	9.49	10.00	9.98	9.69
Number of new grains by coalescence	167	512	554	663	666	651
Number of coalescing initial grains	339	1192	1401	1957	1912	1737
Coalescing volume fraction	3.7%	12.7%	14.8%	20.6%	20.1%	18.2%

**Table 3 materials-13-05834-t003:** Grain coalescence analysis results for the polycrystal with shear-type initial texture.

Shear Strain	0	1	2	4	6	8
Grain number	9776	8531	7784	7132	7361	7630
Average grain size (number-weighted)/μm	8.51	8.72	8.80	8.89	8.88	8.90
Average grain size (volume-weighted)/μm	9.08	10.30	12.40	14.19	13.80	11.50
Number of new grains by coalescence	218	692	775	848	878	919
Number of coalescing initial grains	442	2161	2991	3716	3517	3289
Coalescing volume fraction	4.8%	22.6%	31.3%	38.5%	36.8%	34.1%

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
