# Peer review of "Polycrystal Simulation of Texture-Induced Grain Coarsening during Severe Plastic Deformation"

_materials, 2020, doi:10.3390/ma13245834_

Round 1

Reviewer 1 Report

The manuscript reports the results of a logically structured series of modelling experiments.

Texture induced grain coarsening effect in FCC polycrystals during simple shear, in 3D topology. To construct the 3D polycrystal aggregate the cellular automaton model was used while to assign the orientations Taylor polycrystal plasticity approach was applied. The results point to the role of the initial texture in the texture that develops under SPD. In addition, many new aspects such as grain fragmentation and coalescence, 2D or 3D modelling were illuminated regarding the texture modelling and texture development problem during SPD.

I found only some mistyping and interpretation problems, namely:

row No47: “in number fraction” (?)

row No93: double space between “polycrystal” and “The”

row No. 114: double dot

Figure3: The grains morphology is rather strange for a voxel size of 50x50x50. Surprisingly, this does not cause a problem with the pole figure diagram value.

row No. 237: something is missing after the first “ Taylor”. model? deformation?

Row No. 253: Explain the texture index expression.

Row No. 269: Author contributions is missing.

Row No. 408: Eqs.(A1-A4) instead of Eqs.(1-4)

Author Response

We are thanking you and the reviewers for the useful comments and suggestions on our manuscript. We have modified the manuscript accordingly and are submitting the revised version of our manuscript. All the changes are highlighted in the revised manuscript. 

The following is a point-to-point response to the comments.

--------------------------------------------------------------------

The manuscript reports the results of a logically structured series of modelling experiments.

Texture induced grain coarsening effect in FCC polycrystals during simple shear, in 3D topology. To construct the 3D polycrystal aggregate the cellular automaton model was used while to assign the orientations Taylor polycrystal plasticity approach was applied. The results point to the role of the initial texture in the texture that develops under SPD. In addition, many new aspects such as grain fragmentation and coalescence, 2D or 3D modelling were illuminated regarding the texture modelling and texture development problem during SPD.

I found only some mistyping and interpretation problems, namely:

row No47: “in number fraction” (?)

Response: It is changed to “in number-weighted average”.

row No93: double space between “polycrystal” and “The”

Response: Corrected.

row No. 114: double dot

Response: Corrected.

Figure3: The grains morphology is rather strange for a voxel size of 50x50x50. Surprisingly, this does not cause a problem with the pole figure diagram value.

Response: The grain morphologies seem not realistic in the polycrystal model with 50×50×50 voxels because much less voxel numbers were used to represent the grains than those in the other two models. In average, 50×50×50/10000=12.5 voxels composed a grain in this model, causing the rougher grain boundary and morphology. However, 10000 randomly oriented orientations were assigned to this model. That means the cubic space had 125000 voxels with 10000 grains / orientations. The texture was statistically analyzed based on these orientations. Therefore, the pole figure diagram value is also good for this model.

The concern of using the model with 50×50×50 voxels is that the grain boundaries were rough and the spatial relations between grains may be not precise. This is the reason why different voxel division conditions were used to test the results. And the polycrystal model having 50×50×50 voxels gave a little less coarsening results as compared to the others. This is related to the spatial relations for the model. Therefore, we selected the model with 150×150×150 to develop our results. 

row No. 237: something is missing after the first “ Taylor”. model? deformation?

Response: Sorry for the mistake. The sentence was modified as follows.

“The significant increase in texture intensity is due to the fact that the Taylor plasticity model does not consider…”

Row No. 253: Explain the texture index expression.

Response: The following sentence was added to the revised manuscript.

“The texture index is obtained as the integral of the square of the texture function, which can be used to estimate the sharpness of the texture [24,28].”

Row No. 269: Author contributions is missing.

Answer: Sorry, we do not see the meaning of this comment at this point of the manuscript. The Authors contibutions are now specified at the end of the manuscript.

Row No. 408: Eqs.(A1-A4) instead of Eqs.(1-4)

Response: Corrected.

Reviewer 2 Report

In this work, the authors perform micromechanical simulations with the Taylor type CP model to simulate the grain coarsening induced by the texture evolution. The approach is indeed interesting. However, following points are needed to be clarified. Please address these issues before proceed to the next step.

  • In the introduction, please provide the motivation of using cellular automata (CA) to simulate grain coarsening. Why using CA method is better than other technique? It would be better to strengthen this part.
  • To discretize texture, which is described in terms of a continuous function, a complex mathematical operation is needed. Here, authors only mentioned about the software used. The referee recommends authors to at least describe the principle of the method used in the program to provide a better picture of the approach.
  • The simulation is based on the FEM calculation or what? Please provide more detail in the simulation set up.
  • How the texture-induced grain coarsening mechanism is implemented into the model. Is it only the orientation evolution during the deformation and authors just hope for the grain to be coarser because their crystallographic orientations are rotated in the way that they are align with the neighboring grain? Is that really the case in the reality? What about the grain boundary mobility, grain boundary energy etc.? The assumption made by author is sometimes difficult to digest.
  • To justify the random texture shown in Figure 3, please provide the texture index of these RVEs.
  • What is the support argument that the grain becomes slightly finer after applied shear strain of 4? What is the unit of shear strain? No unit or percent?
  • From the simulation the average grain size increase from around 9 to 10 µm. However, comparing to the experiment reported by Gu et al., it increased significantly. What could be the cause of the mismatch?
  • From line 288-290, please describe how the color code RGB relates to Euler angles?

The referee would suggests a major revision for this manuscript.

Author Response

We are thanking you and the reviewers for the useful comments and suggestions on our manuscript. We have modified the manuscript accordingly and are submitting the revised version of our manuscript. All the changes are highlighted in the revised manuscript. 

The following is a point-to-point response to the comments.

------------------------------------------------------------------

Reviewer #2:

In this work, the authors perform micromechanical simulations with the Taylor type CP model to simulate the grain coarsening induced by the texture evolution. The approach is indeed interesting. However, following points are needed to be clarified. Please address these issues before proceed to the next step.

In the introduction, please provide the motivation of using cellular automata (CA) to simulate grain coarsening. Why using CA method is better than other technique? It would be better to strengthen this part.

Response: Thanks for the comment. 3D polycrystal construction is the base for performing 3D analyses in this work. Several techniques have been developed to generate the 3D polycrystal model. The CA method was used in this work based on the following reasons. Firstly, the CA method models the grain geometry based on voxels, which works well with the discrete spatial detection. Secondly, the CA method is relatively flexible and efficient in polycrystal modeling. Thirdly, the grain boundary morphology generated in CA is convex surface which is more realistic to represent the grain morphology than the flat surface generated by other methods. Also, the grain size distribution follows the Gaussian distribution. For all these reasons, the CA method was used in this work. The following description was added to the manuscript according to the comment.

“Among these models, the CA method models the entire 3D grain geometry based on voxels, which works well with the discrete spatial detection [24]. CA is especially versatile and computationally efficient. Moreover, the 3D grain boundary morphology generated by CA is a convex surface and the grain size distribution follows Gaussian distribution, which are realistic properties for representing a polycrystal.

To discretize texture, which is described in terms of a continuous function, a complex mathematical operation is needed. Here, authors only mentioned about the software used. The referee recommends authors to at least describe the principle of the method used in the program to provide a better picture of the approach.

Response: The following information was added:

“ATEX is using the cumulative ODF based technique for discretization, which is described in detail in Ref. [26].” Ref. [26] is new, we renumbered the others.

The simulation is based on the FEM calculation or what? Please provide more detail in the simulation set up.

Response: Our model is not a finite element model; this is why we did not specify our model as such. We used the cellular automaton model, as it is presented. CA is used frequently for recrystallization simulations. Actually, the simulation is divided into two parts: simulation of the texture evolution and analysis of the microstructure. For the texture evolution part, the Taylor polycrystal plasticity model was used to calculate the orientation evolution without considering the spatial distribution and neighbor effects of grains in the polycrystal. After that, the orientations were output at selected strains to update the orientations of grains in the polycrystal. Then the grain coalescence detection procedure was applied to detect the grain coalescence in the polycrystal model. The main function of the CA is to supply the spatial relations of grains in 3D to check the coalescence after orientation updating. This procedure is described in Fig. 1. While the CA approach is well known, in order to be more clear for non-experts of CA modeling, more details were added to Sections 2.2 and 2.3.

How the texture-induced grain coarsening mechanism is implemented into the model. Is it only the orientation evolution during the deformation and authors just hope for the grain to be coarser because their crystallographic orientations are rotated in the way that they are align with the neighboring grain? Is that really the case in the reality? What about the grain boundary mobility, grain boundary energy etc.? The assumption made by author is sometimes difficult to digest.

Response: Yes, exactly. Nevertheless, our modeling was not based on “hopes”, instead, on rigorous computation. We only aimed to see the extent and effect of grain coarsening due to texture evolution, as it is stated in the title and abstract. We did not talk about grain boundary mobility, as this is not a dynamic recrystallization study, it is solely texture evolution. Dynamic recrystallization calculations are usually done with the CA model, but we did not talk about it as it was not the subject of our study. We believe that with the new modifications the revised manuscript is clear, without misunderstanding. We agree that more simultaneous phenomena should be examined at the same time for approaching the real material behavior, we are working on them, like grain fragmentation due to orientation gradients within grains. Those results will be published soon. In this work, we wanted to isolate the texture induced effect, in order to get quantitative estimations on its importance.  

To justify the random texture shown in Figure 3, please provide the texture index of these RVEs.

Response: Thanks for the suggestion. The texture indexes were added to figure 3.

What is the support argument that the grain becomes slightly finer after applied shear strain of 4? What is the unit of shear strain? No unit or percent?

Response: The supporting argument is the result itself. Nevertheless, one can argue using the very specific nature of the rotation field of grains within orientation space. That is what we did. We added more about it in the Discussion part; when only orientation change is looked at, the main reason for grain refinement is the convergent and divergent rotation fields around the ideal shear orientations. When the grain orientations cross from the convergent side to the divergent part of the rotation field, some previously coarsening grains will split again since their different parts rotate to different orientations, as shown in Fig. 11. This implies that there exist some limiting values for the texture induced grain coarsening. This is what we were guessing only before starting our simulations; it turned out to be true.

 Concerning the shear strain, strain in mechanics has no unit. Materials science researchers use percentage in some cases; they are forgiven by the mechanics community.

From the simulation the average grain size increase from around 9 to 10 µm. However, comparing to the experiment reported by Gu et al., it increased significantly. What could be the cause of the mismatch?

Response: We have given the response in the paper, we recall: “Clearly, the grains have many more number of neighbors in 3D, moreover, the coalescence can also ‘propagate’ in 3D, while its propagation is very limited in 2D. Therefore, the coalescing volume fraction is much higher in 3D than in 2D.”

From line 288-290, please describe how the color code RGB relates to Euler angles?

Response: Thanks for pointing out this. We checked the manual of Dream 3D and added the color legend in Fig. 13.

The referee would suggest a major revision for this manuscript.

Reviewer 3 Report

The main question in the present work is focused on the modelling of the grain-coarsening effect induced by the evolution of the crystallographic texture.

In my opinion, the presented modeling results are interesting and show important investigates of the texture induced grain coarsening effect in FCC polycrystals during simple shear, in 3D topology.

The presented results are not very original and unique, but they make a significant contribution to the discussed issues.

A lot of models were developed to simulate the grain fragmentation during SPD process. These models show reasonable agreements on the grain size variations and texture evolution.  However, very less attention is paid on the phenomenon of possible grain coarsening during SPD. The main subject of the present manuscript is to establish 3D polycrystalline aggregates using a 3D CA method. The texture evolution was simulated for simple shear deformation mode using the Taylor polycrystal model. The coalescence of grains induced by the texture evolution was tracked by detecting the disorientations between neighbor grains. The material volume that was concerned by grain coalescence depended on the initial texture; it was the most significant for an initial shear texture, where it went up to near 40% in a shear strain of about 4.

I believe that the manuscript is written in the correct language.

The presented manuscript is easy to read and has an appropriate structure.

In the present work the grain-coarsening effect induced by the evolution of the crystallographic texture was modelled. The approach consisted in establishing a 3D polycrystal aggregate with large number of grains (10000). Each grain was composed of voxels (typically several hundred) and its orientation evolution was followed by the Taylor crystal plasticity model. Simple shear deformation was considered as this is the most frequent in SPD experiments. The coalescence events  of neighbor grains were detected when the disorientations between neighboring voxels of adjacent grains became less than 5°. In my opinion, the conclusions are consistent with presented arguments.

Author Response

We are thanking you and the reviewers for the useful comments and suggestions on our manuscript. We have modified the manuscript accordingly and are submitting the revised version of our manuscript. All the changes are highlighted in the revised manuscript. 

The following is a point-to-point response to the comments.

------------------------------------------------------------------

Reviewer #3:

The main question in the present work is focused on the modelling of the grain-coarsening effect induced by the evolution of the crystallographic texture.

In my opinion, the presented modeling results are interesting and show important investigates of the texture induced grain coarsening effect in FCC polycrystals during simple shear, in 3D topology.

The presented results are not very original and unique, but they make a significant contribution to the discussed issues.

A lot of models were developed to simulate the grain fragmentation during SPD process. These models show reasonable agreements on the grain size variations and texture evolution.  However, very less attention is paid on the phenomenon of possible grain coarsening during SPD. The main subject of the present manuscript is to establish 3D polycrystalline aggregates using a 3D CA method. The texture evolution was simulated for simple shear deformation mode using the Taylor polycrystal model. The coalescence of grains induced by the texture evolution was tracked by detecting the disorientations between neighbor grains. The material volume that was concerned by grain coalescence depended on the initial texture; it was the most significant for an initial shear texture, where it went up to near 40% in a shear strain of about 4.

I believe that the manuscript is written in the correct language.

The presented manuscript is easy to read and has an appropriate structure.

In the present work the grain-coarsening effect induced by the evolution of the crystallographic texture was modelled. The approach consisted in establishing a 3D polycrystal aggregate with large number of grains (10000). Each grain was composed of voxels (typically several hundred) and its orientation evolution was followed by the Taylor crystal plasticity model. Simple shear deformation was considered as this is the most frequent in SPD experiments. The coalescence events of neighbor grains were detected when the disorientations between neighboring voxels of adjacent grains became less than 5°. In my opinion, the conclusions are consistent with presented arguments.

Response: Thanks for your comments. We sincerely hope this paper can make a good contribution to the field of SPD research. 

Reviewer 4 Report

Submitted manuscript entitled “Polycrystal simulation of texture induced grain coarsening during severe plastic deformation" described in details the texture induced coarsening effect in SPD processing which have to be taken into account in the grain fragmentation process. Topic is interesting and I recommend it to publication. I think paper is written in good enough language, but maybe English language verification should be done by the right person for English spelling and grammar. Technique, technology and research methods used in the work are adequate. Methods and obtained results prove founded thesis and show originality of the manuscript.

Some small revision of paper is needed.

Figure 2: I am wondering about the description of the 0Y axis in both of these figures? Figure 2a and 2b, there is respectively Volume fraction and Number fraction, (what is unit), and in the capture of the figure there is written: a) Number fraction and b) Volume fraction. Should be a) Volume fraction and b) Number fraction

Author Response

We are thanking you and the reviewers for the useful comments and suggestions on our manuscript. We have modified the manuscript accordingly and are submitting the revised version of our manuscript. All the changes are highlighted in the revised manuscript. 

The following is a point-to-point response to the comments.

-----------------------------------------------------------------

Reviewer #4:

Submitted manuscript entitled “Polycrystal simulation of texture induced grain coarsening during severe plastic deformation" described in details the texture induced coarsening effect in SPD processing which have to be taken into account in the grain fragmentation process. Topic is interesting and I recommend it to publication. I think paper is written in good enough language, but maybe English language verification should be done by the right person for English spelling and grammar. Technique, technology and research methods used in the work are adequate. Methods and obtained results prove founded thesis and show originality of the manuscript.

Some small revision of paper is needed.

Figure 2: I am wondering about the description of the 0Y axis in both of these figures? Figure 2a and 2b, there is respectively Volume fraction and Number fraction, (what is unit), and in the capture of the figure there is written: a) Number fraction and b) Volume fraction. Should be a) Volume fraction and b) Number fraction

Response: Thanks for your comments. The capture of Fig. 2 was corrected. The grain size distribution of the 3D polycrystal model was analyzed using different statistical methods in volume fraction and number fraction (no unit for this). Besides, we modified the whole manuscript carefully and hope the language is now acceptable for publication.

Thanks to all Referees and the Editor. We are looking forward to the acceptability of the revised manuscript.  

Round 2

Reviewer 2 Report

The authors has revised and clarified all points raised by the referee. 

The referee now suggests to accept the manuscript in the current form.